# Translation of non-standard codon nucleotides reveals minimal requirements for codon-anticodon interactions

Thomas Philipp Hoernes[1], Klaus Faserl[2], Michael Andreas Juen[3], Johannes Kremser[3], Catherina Gasser[3], Elisabeth Fuchs[3], Xinying Shi[4], Aaron Siewert[5], Herbert Lindner[2], Christoph Kreutz[3], Ronald Micura[3], Simpson Joseph[4], Claudia Höbartner[5], Eric Westhof[6], Alexander Hüttenhofer[1] & Matthias David Erlacher[1]

The precise interplay between the mRNA codon and the tRNA anticodon is crucial for ensuring efficient and accurate translation by the ribosome. The insertion of RNA nucleobase derivatives in the mRNA allowed us to modulate the stability of the codon-anticodon interaction in the decoding site of bacterial and eukaryotic ribosomes, allowing an in-depth analysis of codon recognition. We found the hydrogen bond between the $N^1$ of purines and the $N^3$ of pyrimidines to be sufficient for decoding of the first two codon nucleotides, whereas adequate stacking between the RNA bases is critical at the wobble position. Inosine, found in eukaryotic mRNAs, is an important example of destabilization of the codon-anticodon interaction. Whereas single inosines are efficiently translated, multiple inosines, e.g., in the serotonin receptor 5-HT$_{2C}$ mRNA, inhibit translation. Thus, our results indicate that despite the robustness of the decoding process, its tolerance toward the weakening of codon-anticodon interactions is limited.

[1] Division of Genomics and RNomics, Biocenter, Medical University of Innsbruck, 6020 Innsbruck, Austria. [2] Division of Clinical Biochemistry, Biocenter, Medical University of Innsbruck, 6020 Innsbruck, Austria. [3] Institute of Organic Chemistry and Center for Molecular Biosciences (CMBI), University of Innsbruck, 6020 Innsbruck, Austria. [4] Department of Chemistry and Biochemistry, University of California at San Diego, 9500 Gilman Drive, La Jolla, CA 92093-0314, USA. [5] Institute of Organic Chemistry, University of Würzburg, Am Hubland, 97074 Würzburg, Germany. [6] Architecture and Reactivity of RNA, Institute of Molecular and Cellular Biology of the CNRS UPR9002/University of Strasbourg, Strasbourg 67084, France. Correspondence and requests for materials should be addressed to M.D.E. (email: matthias.erlacher@i-med.ac.at)

In RNA, Watson–Crick (W–C) base pairing is ubiquitous but is only one of numerous possible interactions that can be formed due to the single-stranded nature of RNA[1–3]. This structural versatility enables single-stranded RNA not only to contain and transport simple sequence information in the form of messenger RNAs (mRNAs) but also to execute enzymatic functions as ribozymes[4–6]. During protein synthesis, both the structural variability and the sequence information of RNA are absolutely essential. Transfer RNAs (tRNAs) and ribosomal RNAs (rRNAs) form characteristic and elaborate tertiary structures enabling the optimized and fine-tuned translation of mRNAs into proteins[7–9]. However, the very basis for the decoding of mRNA sequences during protein synthesis is W–C base pairing[10,11]. At the ribosomal A site, the mRNA codon is presented to the incoming tRNA anticodon, thereby forming W–C interactions. Whereas W–C base pairing at the first and second codon nucleotide is strictly required, the conformation at the third, the so-called wobble position, is structurally more flexible[12–14].

Extensive structural, physico-chemical, and kinetic studies of the decoding process have revealed that an integrative interaction network between mRNA, tRNA, and rRNA ensures efficient and accurate translation[15–17]. Additional factors have been identified that modulate the decoding process by impacting the quality of the codon–anticodon interaction. A significant and essential influence on decoding derives from RNA modifications[18–21]. tRNAs, in particular, contain many post-transcriptionally modified nucleotides. Also, rRNAs and mRNAs harbor numerous base and ribose modifications, implying that they may play an important role during protein synthesis[22–25]. Their functions, however, are still largely unknown.

A well characterized non-standard nucleoside in mRNAs is inosine (I)[26]. I is a result of a hydrolytic deamination of adenosine by a family of proteins called ADARs[27]. This editing event leads to a switch from a hydrogen donor (amino group) to a hydrogen acceptor (carbonyl oxygen) at position 6, generating a W–C edge reminiscent of guanosine (G), thereby altering the genetic information through the preferential base pairing of I with cytosine (C)[27]. In addition, the I–C interaction is less stable than that of G–C, mainly due to the loss of one hydrogen bond (H-bond)[28]. Since inosine has also been revealed in coding sequences (CDSs) of mRNAs, it is remarkable that, so far, inosine has not been observed to impair protein synthesis. This potentially implies that the number of H-bonds between codon and anticodon is less critical during translation than previously assumed or that a loss in stability of the codon–anticodon interaction can be compensated for by other means[29,30].

Decoding of mRNAs has been extensively studied during the last decades. Recently, complementary packing and hydrophobic forces have been demonstrated to be crucial for decoding[31]; however, the contribution of single H-bonds to the codon–anticodon interaction has not yet been systematically addressed. By specific insertions of various non-natural modifications into mRNA codons (Fig. 1), we intended to define the limits for the stability of the W–C interaction in the ribosomal decoding site during protein synthesis. Strikingly, in bacteria as well as in eukaryotes, the number of H-bonds at single codon positions only marginally affected protein synthesis. Thus, it is not the number of H-bonds, but rather the contact interactions that maintain the overall geometry and shape of the base pair that are critical for translation.

## Results

**Non-standard nucleotides as tools to investigate decoding.** To manipulate W–C interactions at the decoding site of the ribosome, a variety of non-natural RNA nucleobase derivatives were introduced site-specifically into reporter mRNAs (Fig. 1). This was achieved by employing chemically synthesized oligonucleotides harboring the desired base modification. Due to the length limitations of these oligonucleotides, a splinted ligation was carried out covalently linking the modified 3′-fragment to a 5′-transcript, resulting in a full-length mRNA[32,33]. The ligation products were purified and subsequently served as templates for translation. To investigate bacterial and eukaryotic translation processes, the recombinant PURExpress system (NEB)[34] and HEK293T cells were employed, respectively[32,33].

**Defining the basic rules of the codon–anticodon interactions.** To define the boundaries for efficient decoding, the codon–anticodon interactions were drastically weakened by inserting base modifications within a GGG (Gly) codon[15]. Thereby, the codon context is strong in terms of hydrogen bonding, and the respective tRNA$^{Gly}$ carries a comparably low number of modifications, thus reducing the contribution of tRNA modifications to decoding[15,18]. Initially, benzimidazole (Benz) and a ribose abasic site (Rab) were site-specifically introduced into this codon. Benzimidazole (Fig. 1f) cannot form H-bonds with a pyrimidine base due to the absence of the N$^1$ and the 6-amino group, assuming that a W–C geometry is formed (Fig. 1a, f). Due to the missing heterocycle of the purine or pyrimidine base, Rab-sites do not provide any stabilization of the decoding site via base stacking, which further weakens the codon–anticodon interaction (Fig. 1f). Not unexpectedly, in both bacterial and eukaryotic systems the introduction of Rab-sites did not allow translation of the modified mRNA, independent of its position within the GGG codon (Fig. 2a, b; Supplementary Fig. 1). Similarly, mRNAs containing Benz were not translated when it was located at the first or second codon position. However, at the wobble position, Benz allowed for protein synthesis comparable to that of unmodified mRNAs (Fig. 2a, b).

A less drastic modification, in respect to exocyclic groups potentially participating in H-bonding, is pyridone (Py) (Fig. 1f) in a UUU codon. In a W–C geometry, however, Py should not favor base pairing with A because of the close contact between the amino group of A and the C$^3$-H of Py. W–C base pairing with G is also not favorable due to a repulsion of the C$^3$-H by N$^1$-H (Supplementary Fig. 2a). However, Py potentially forms a standard wobble pair with G (Supplementary Fig. 2b). As anticipated, Py could not be translated at the first or second codon position, reflected also in a decreased binding of tRNA ternary complexes (Supplementary Fig. 3 and Supplementary Table 1). The respective peptide products were only detected when Py was located at the wobble position (Fig. 2c, d; Supplementary Fig. 1). Upon introducing zebularine (Ze) in UUU codons, thereby creating a second H-bond (Fig. 1b), efficient translation of the modified mRNA was observed in the PURExpress system and in HEK293T cells (Fig. 2c, d).

Modifications of purine bases were tested within a weak codon (AAA)[15]. It has been previously postulated that tRNA sequences, structures, and modifications evolved to compensate for differences in the stability of the codon–anticodon interaction, allowing a uniform selection by the ribosome[15,30,35]. Therefore, a weak codon might be potentially less sensitive to the reduction of W–C interactions.

Indeed, mRNAs containing purine (P) that only formed one H-bond per base pair (Fig. 1c) were efficiently translated in bacteria (Fig. 2e, f; Supplementary Fig. 1), despite a drastic destabilization of the codon–anticodon interaction (Supplementary Fig. 4). Additionally, in HEK293T cells, P was tolerated at the first and third codon position but, remarkably, did not allow for efficient translation at the second nucleotide. Strikingly, CPC

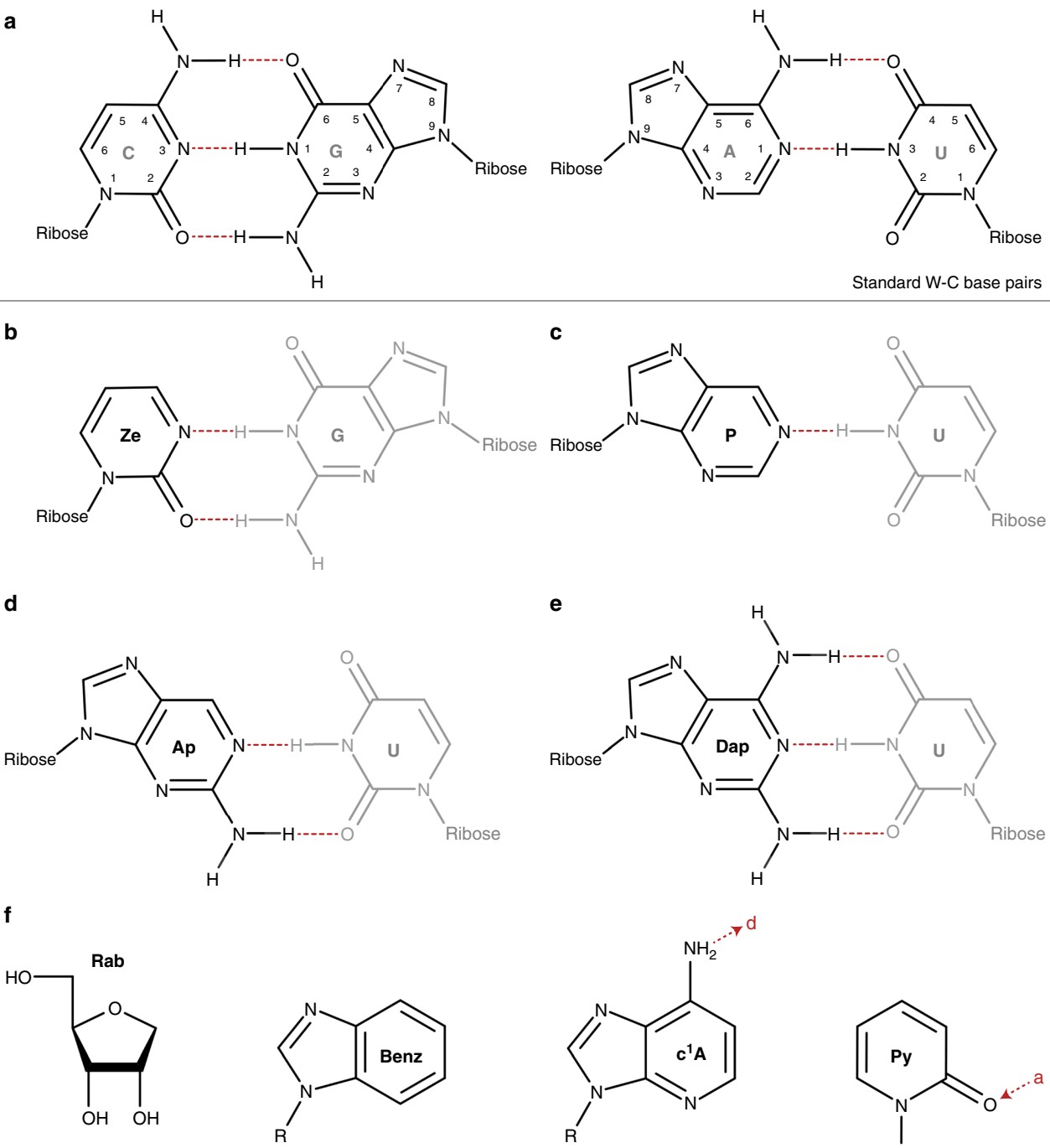

**Fig. 1** Interactions between standard and modified bases. **a** Standard C–G and A–U base pairs and the H-bonds formed are depicted. **b**–**e** Modified bases employed in this study and their potential binding partners are shown in W–C-like geometry. The modified and the standard nucleobases are depicted in black and gray, respectively. **f** Chemical structures of modified bases that either cannot form H-base pairs or depend on an exocyclic group. H-bonds are depicted in red (Ze zebularine, P purine, Ap 2-aminopurine, Dap 2,6-diaminopurine, Rab ribose abasic, Benz benzimidazol, c[1]A 1-deazaadenosine, Py 2-pridone, d H-bond donor, a H-bond acceptor)

encoding for Arg was efficiently translated (Supplementary Fig. 5). To shed more light on this observation and to examine not only the number but also the position of the H-bonds, 1-deazaadenosine (c[1]A) was investigated (Fig. 1f). In analogy to P, c[1]A can potentially only form a single H-bond but, in this case, through the 6-amino group and not through the N[1] (Supplementary Fig. 2c). c[1]A was introduced separately at every position within an AAA codon. In such codons, c[1]A did not enable protein synthesis at the first or the second position (Fig. 2e, f).

This can be rationalized not only by the impact of the position of the single H-bond but also by the close proximity of the C[1]-H and N[3]-H of the c[1]A and the U, respectively (Supplementary Fig. 2c). Consistently, the c[1]A–U interaction was drastically destabilized in the minimal codon–anticodon model system, decreasing the $T_m$ below the detection limit (Supplementary Fig. 4). Unexpectedly, the third codon position c[1]A could also not be decoded, presumably, because a proper wobble base pair could not be formed (Fig. 2e, f; Supplementary Fig. 2d).

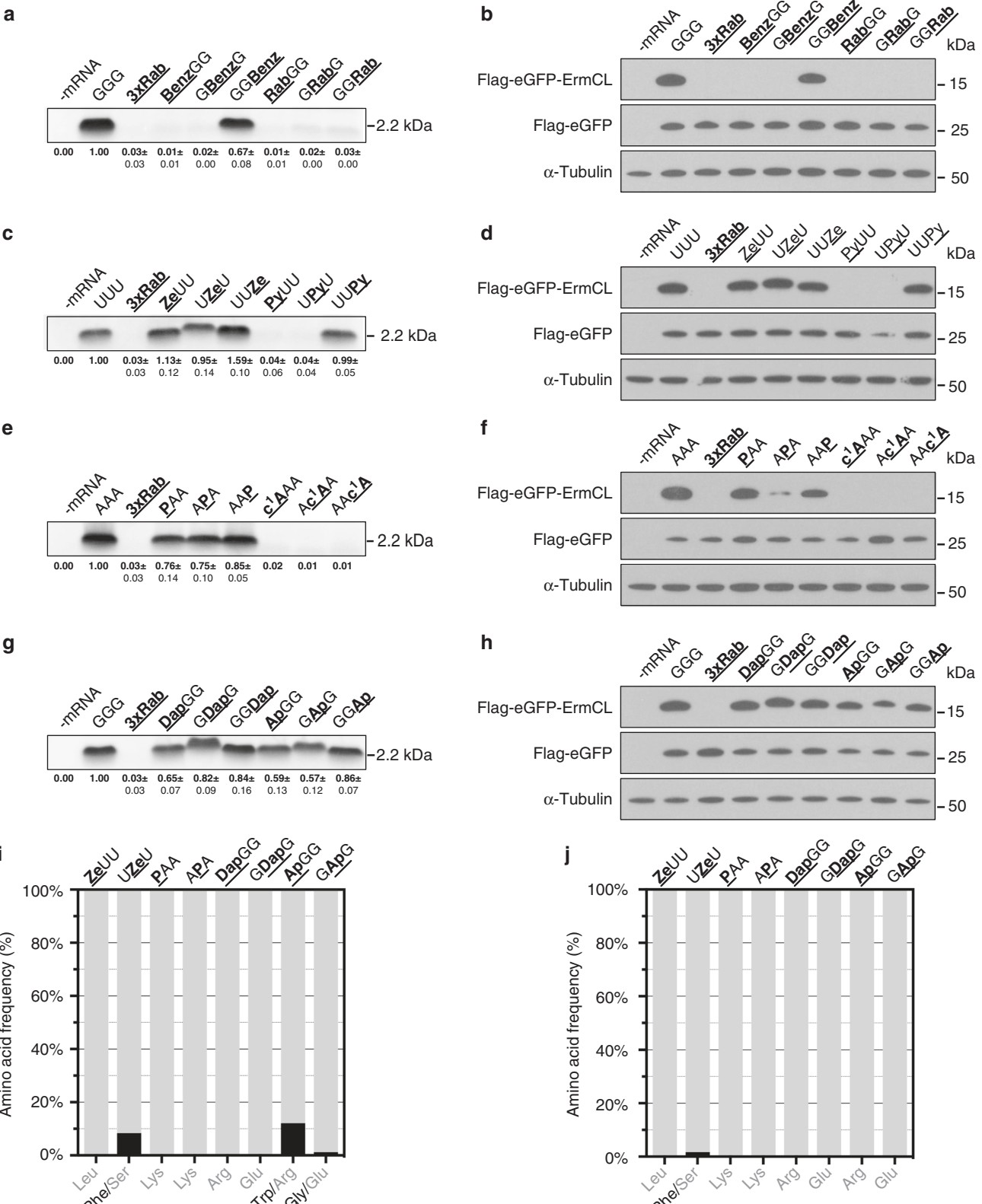

**Fig. 2** Translation of mRNAs carrying non-standard nucleotides. Left panel **a**, **c**, **e**, **g**: Autoradiography of ErmCL peptides synthesized using the bacterial in vitro translation system. Right panel **b**, **d**, **f**, **h**: Western blot analyses of modified mRNAs translated in HEK293T cells. An unmodified eGFP mRNA and α-tubulin served as an internal transfection control and as a loading control, respectively. Amino acids incorporated at the respective positions of peptides purified from **i** the *E. coli* translation system and **j** HEK293T cells were analyzed by mass spectrometry. Error bars depict the SDs from the mean of at least three independent experiments

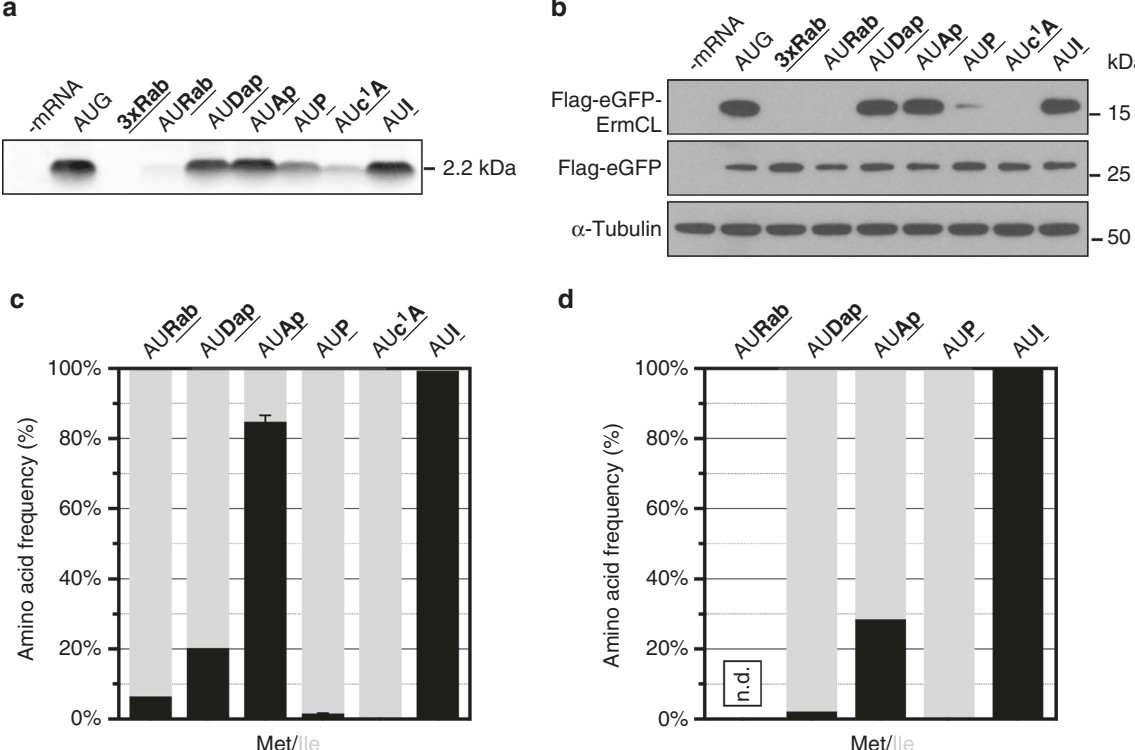

**Fig. 3** Effects of modified nucleotides in the wobble position. Modified nucleotides were introduced in the wobble position and peptides were analyzed after translation in **a** an *E. coli* in vitro translation system using autoradiography or **b** in HEK293T cells employing western blot analysis. Purified translation products from the **c** *E. coli* system and **d** HEK293T cells were analyzed by mass spectrometry (n.d. not determined)

Since we observed that not only the number but also the positions of the formed H-bonds were important, 2-aminopurine (Ap) and 2,6-diaminopurine (Dap) (Fig. 1d, e) were incorporated to evaluate the importance of the exocyclic group for base pairing. In the bacterial and eukaryotic translation systems, both modifications enabled efficient protein synthesis (Fig. 2g, h; Supplementary Fig. 1). In the case of Dap, even the increased number of H-bonds did not significantly alter the translation process (Fig. 2g, h). P, Ap, and Dap had similar translation levels in both bacteria and eukaryotes.

**Codon–anticodon interactions that define sense codons.** In addition to the protein products that are formed in dependence of the modified codons, the interpretation of these RNA nucleobase derivatives by the ribosomal decoding site was also determined. For each modification, the translated peptides from the bacterial and eukaryotic systems were purified and analyzed by mass spectrometry (MS) (Fig. 2i, j; Supplementary Tables 2–4). In respect to the pyridine modifications, Ze was decoded as C since ZeUU and UZeU codons resulted in the incorporation of leucine (Leu) and serine (Ser), respectively. In bacteria, UZeU was also partly decoded (~8%) as a phenylalanine (Phe) codon, indicating that Ze can also base pair with A to a limited extent (Fig. 2i; Supplementary Table 2). Although the base pair Ze–A is also observed in HEK293T cells, it is less abundant than in bacteria (Fig. 2j; Supplementary Tables 3 and 4).

P within AAA codons was always translated as lysine (Lys) independent of its position within the codon and the translation system (Fig. 2i, j; Supplementary Tables 2 and 3). By placing an exocyclic amino group at position 2, a shift in decoding from A to G was assumed, since the 2-amino group is also present in G. In eukaryotes, however, Ap was exclusively read as A at the first two

codon positions (Fig. 2j; Supplementary Table 3). In bacteria, ApGG led to the incorporation of tryptophan (Trp), which was a result of decoding ApGG as UGG (Fig. 2i; Supplementary Table 2). This is remarkable, since the Trp incorporation would require a purine–purine base pair within the codon–anticodon interaction. Ap at the second nucleotide of the codon led to low levels of Gly in addition to Glu, caused by reading Ap as G (Fig. 2i; Supplementary Table 2). Placing a 6-amino group in addition to the 2-amino group into the mRNA, by integrating Dap, led to an unambiguous base pairing of Dap with U in both tested translation systems (Fig. 2i, j; Supplementary Tables 2 and 3).

**Decoding of non-standard nucleotides at the wobble position.** Due to the degenerate nature of the genetic code, addressing the decoding of modified bases at the wobble position is limited to the AUG codon. AUG is the only codon that enables the differentiation of whether the ribosome interprets the modified purine base as A or G, since AUG decodes for methionine (Met) and AUA/C/G for isoleucine (Ile).

The codon AURab was not efficiently translated (Fig. 3a, b). Nevertheless, the bacterial translation system provided sufficient amounts of the peptide for MS analysis, revealing that the majority of peptides contained Ile and only a fraction of the peptides contained Met (Fig. 3a, c; Supplementary Tables 2 and 4), possibly reflecting the abundance of the respective tRNAs[36]. In accordance with the results obtained when P was positioned at the first two codon nucleotides, this base derivative was decoded almost exclusively as an A in the AUP codon (Fig. 3c, d; Supplementary Table 2–4). Remarkably, the AUP codon was not as efficiently translated as the AAP codon (Figs. 3a, b and 2e, f). Furthermore, the introduction of c¹A at the third codon position

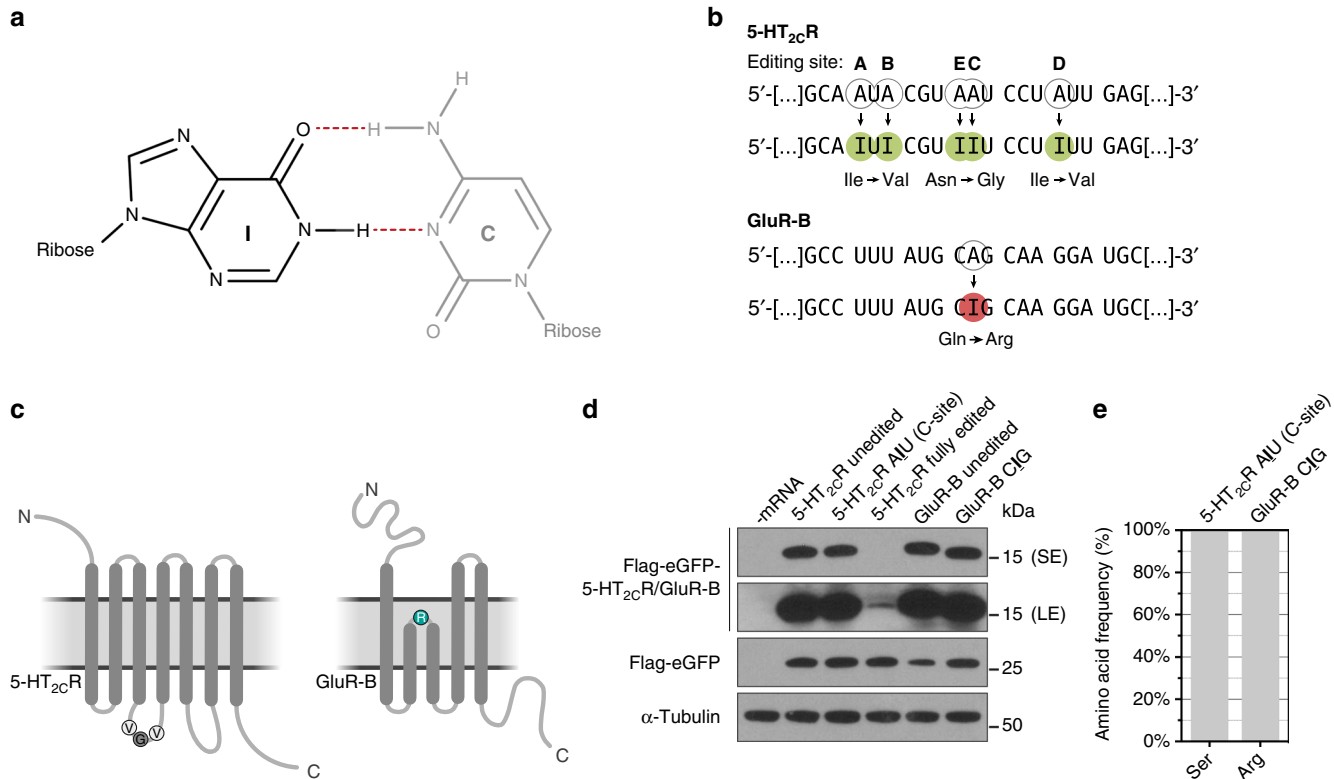

**Fig. 4** Inosine is a non-standard nucleotide present in naturally occurring mRNAs. **a** The I–C base pair forms two hydrogen bonds within the W–C geometry. **b** The mRNAs of 5-HT$_{2C}$R and GluR-B are subject to A-to-I editing, thereby altering their genetic code. The editing sites are encircled. **c** Schematic representation of 5-HT$_{2C}$R and GluR-B proteins with the affected amino acids highlighted. **d** The I-containing mRNAs were transfected into HEK293T cells and the translation products were analyzed by western blotting (SE short exposure, LE long exposure). **e** The identity of the incorporated amino acids was determined by MS

of the Lys codon did not result in any translation product, whereas sufficient amounts of peptide could be detected for AUc$^1$A, which was decoded as isoleucine (Fig. 3a, c; Supplementary Table 2).

At the wobble position, the relocation of the exocyclic amino group from position 6 to 2 indeed increased the incorporation of Met, caused by Ap being decoded as G (Fig. 3c, d; Supplementary Tables 2 and 3). This effect was more pronounced in the bacterial than in the eukaryotic translation system. The simultaneous presence of 2- and 6-amino groups reduced the incorporation of Met, indicating a compensatory effect of the 6-amino group in the presence of the 2-amino group (Fig. 3c, d; Supplementary Tables 2 and 3).

The discrimination between AUA and AUG codons is different between bacteria and eukaryotes[37]. In bacteria, AUA is decoded by a tRNA carrying a 2-lysyl-C34 and the AUG codon by a tRNA containing a 2-acetyl-C34. In eukaryotes, AUG is also decoded by a 2-acetyl-C34 tRNA, but the AUA codon by two different tRNAs, one carrying I34 and the other one with a modified U34 base (that can be a pseudouridine)[18]. The present data show that the eukaryotic system is less permissive than the bacterial system. Interestingly, Ap, and Dap to a lesser extent and only in bacteria, pair favorably with 2-acetyl-C34, which would imply a wobble pair where the pyrimidine is pushed in the minor groove instead of the major groove[12]. Otherwise, as described above, Dap and Ap behave as As.

**Inosine affects the genetic variability and tRNA binding.** A naturally occurring example of an altered number of H-bonds during decoding is the translation of I-containing codons. The I–C base pair forms only two H-bonds, whereas the canonical G–C base pair exhibits three H-bonds (Fig. 4a). The conversion of adenosines to inosines, designated as A-to-I editing, is the most prevalent form of RNA editing, with possibly more than 100 million potential modification sites in the human transcriptome[38].

One of the most prominent and best-studied examples of A-to-I editing is the serotonin 5-HT$_{2C}$ receptor (5-HT$_{2C}$R) mRNA. In total, five editing positions on exon V, designated site A to site E, have been reported (Fig. 4b, c); editing at these sites affects G-protein coupling and, consequently, the receptor's activity[39–41]. Another transcript that is almost quantitatively edited is the glutamate receptor GluR-B mRNA expressed in the brain, where at the protein level, the editing event leads to the incorporation of arginine instead of glutamine (Fig. 4b, c). This amino acid exchange is associated with the altered calcium permeability of the receptor[42]. We incorporated single inosines in the mRNA contexts of either 5-HT$_{2C}$R or GluR-B and investigated their influence on eukaryotic translation. We found that single inosines did not affect the yield of the translated peptide product (Fig. 4d). As expected, the inosines were exclusively decoded as G, resulting in an amino acid change from Gln to Arg and from Asn to Ser in the case of the modified 5-HT$_{2C}$R and GluR-B mRNAs, respectively (Fig. 4e; Supplementary Table 3). However, the simultaneous presence of five inosines within the 5-HT$_{2C}$R mRNA completely abolished its translation (Fig. 4d).

Therefore, we studied the effect of I on the stability of the codon–anticodon interaction in both the absence and the presence of the ribosome (Fig. 5). Initially, we employed a minimal codon–anticodon system in solution and measured the

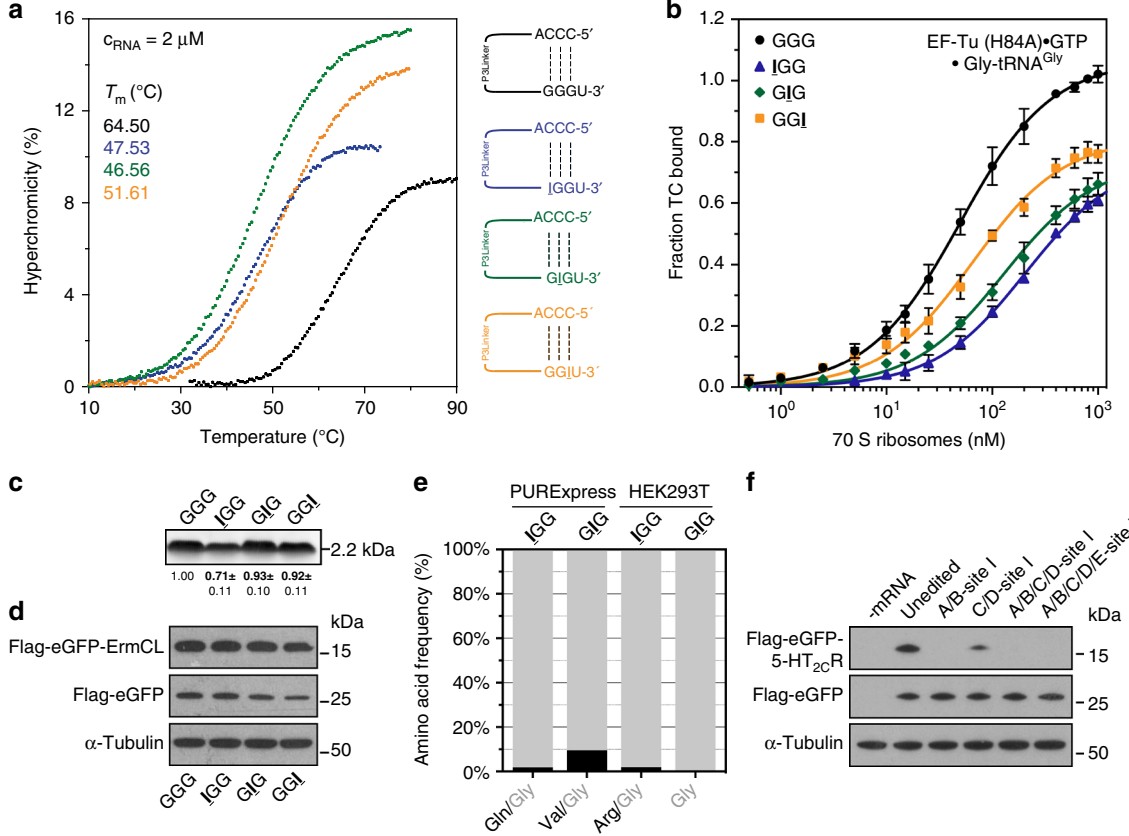

**Fig. 5** The influence of inosine on the stability of the codon–anticodon interaction. **a** Employing a minimal codon–anticodon model, the impact of a single inosine on the stability of the W–C interaction was determined. **b** The observed position-dependent effects on codon–anticodon interaction could be reproduced using an EF-Tu ternary complex (TC)-binding assay. mRNAs carrying single inosines were translated in **c** an *E. coli* translation system and **d** in HEK293T cells. **e** The translation products were purified and qualitatively assessed by mass spectrometry. **f** An additive inhibitory effect of two or more inosines within the 5-HT$_{2C}$R mRNA can be observed. Error bars depict the SDs from the mean of at least three independent experiments

interaction strength of inosines within a glycine codon (GGG) using CCC as an anticodon (Fig. 5a)[43,44]. In such constructs, the stability of the codon–anticodon interaction changed depending on the position of inosine. The respective $T_m$ values (IGG: 47.5 °C, GIG: 46.6 °C, GGI: 51.6 °C) indicated that I–C base pairs, especially in the first and the second codon positions, are significantly less stable compared to an unmodified codon (GGG: 64.5 °C).

We then asked whether this is also the case when an inosine-containing codon is present in the ribosomal A site. To enable an analysis of initial selection at inosine-encoding codons, we measured binding rates of EF–Tu ternary complexes in the A/T state of the 70S ribosome[45]. Due to the limited availability of purified *E. coli* tRNAs, we synthesized a fully modified tRNA$^{Gly}$ (Supplementary Fig. 6) that was subsequently radioactively labeled and charged. Ternary complexes of the Gly-[$^{32}$P]-tRNA$^{Gly}$ with EF–Tu (His84Ala) and GTP were formed (Supplementary Fig. 7). The subsequent filter-binding experiments were performed using varying concentrations of the ribosome complexes (Fig. 5b; Supplementary Table 1). The equilibrium dissociation constant ($K_D$, depicted as $K_D$ ± standard deviation from the mean of at least three independent experiments) with an unmodified GGG codon in the A site was found to be 49.1 ± 2.0 nM. Binding of mRNAs with IGG or GIG codons to the ribosome increased $K_D$ values compared to GGG more than 4-fold or 2.6-fold, respectively (IGG: $K_D$ = 205.3 ± 8.9 nM and GIG: $K_D$ = 127.9 ± 7.8 nM; Fig. 5b; Supplementary Table 1). GGI codons only marginally increased $K_D$ values to 64.8 ± 3.5

nM. Thus, the binding data are in line with the results obtained in solution and indicate that inosines in the A site moderately but significantly interfere with binding of EF–Tu ternary complexes. In particular, inosines in the first and the second codon positions decreased the stability of the codon–anticodon interaction.

Next, we investigated whether this position-dependent effect of inosine could be reproduced in an authentic translation setting. We analyzed the translation of IGG, GIG, and GGI-encoding mRNAs with the PURExpress translation system (Fig. 5c) and HEK293T cells (Fig. 5d). In bacteria, inosine in the first codon position reduced translation rates by approximately 30% (Supplementary Fig. 1); this effect could not be observed in mammalian cells, however. Furthermore, we determined the amino acids that were incorporated at the respective positions of the peptides (Fig. 5e; Supplementary Tables 2 and 3). In both bacterial and eukaryotic translation systems, miscoding was observed at low levels when inosine was decoded in the first codon position: the IGG codon was read 0.4% and 1.9% of the time as AGG (Arg) (or CGN) in bacteria and eukaryotes, respectively. Further, in bacteria, IGG was read 1.4% of the time as CAG (Gln) (Fig. 5e; Supplementary Table 4). GIG codons were also decoded as Val (GUG) in bacteria. Thus, especially in bacterial translations, purine–purine interactions (either G–I or A–I) are observed to a minor extent. Inosine in the wobble position of an AUG codon was almost exclusively decoded as guanosine in bacteria and eukaryotes (Fig. 3c, d; Supplementary Tables 2 and 3).

Although single inosines impair binding of ternary complexes in a codon position-dependent manner, ribosomal elongation was only moderately affected. However, a simultaneous incorporation of two inosines within one codon (A- and B-sites of the 5-HT$_{2C}$R mRNA) completely abolished translation of the edited mRNA (Figs. 4b and 5f). Even inosines within two distinct codons (C- and D-sites of the 5-HT$_{2C}$R mRNA) drastically reduced translation rates, indicating that inosines exhibit an additive inhibitory effect, at least when present in close proximity to each other.

## Discussion

The codon–anticodon interaction is without doubt one of the most crucial interactions in molecular biology. Decades of research have evaluated numerous aspects and factors contributing to the speed and accuracy of the decoding process during protein synthesis. Although these efforts have led to a detailed picture of the translation process, the contribution of single H-bonds between the W–C edges has not been biochemically investigated so far[16,46,47]. Since the formation of stable complexes between the codon and anticodon is apparently crucial for decoding, we investigated the contributions of H-bonds in respect to their number and positions within the W–C geometry by inserting non-natural RNA nucleobase derivatives in the mRNA codons. We find that the translation process in bacteria and eukaryotes is astonishingly robust against the loss of single H-bonds and resulting in the destabilization of the W–C base pairs (Supplementary Table 4). Translation of the respective codon is only modestly impaired, when H-bonds between the purine-N$^1$ and the pyrimidine-N$^3$ at the first two codon positions are formed. Thus, in the cases of pyridone or c$^1$A, the single H-bond is at a different location and translation efficiencies are drastically reduced. The only exception was the translation of single purines within an AAA codon (Lys) in HEK293T cells. Whereas in E. coli, the codon APA could be efficiently decoded, it did not serve as an efficient template for translation in HEK293T cells. This may be explained by the presence of tRNA modifications in human tRNA$_3$$^{Lys}$$_{UUU}$ that potentially destabilize the interaction with APA[48], since this effect is not observed in the case of other codons (Supplementary Fig. 5).

Another difference between translation in bacteria and eukaryotes is the accuracy in the decoding of certain RNA nucleobase derivatives at defined codon positions. Generally, bacterial translation seems to be less restrictive towards the translation of modified codon nucleotides, as observed for the codons UZeU, ApGG, and GIG. In case of ApGG, the bacterial ribosome incorporates more than 10% of Trp into the peptide (Fig. 2i). However, this requires the formation of a purine–purine base pair at the first codon position. Another purine–purine pair was found when translating the GIG codon, but, again, only in the bacterial system. In the case of UZeU, the bacterial translation system incorporates Phe instead of Ser (Fig. 2i). Ze, in all other cases, was read as a C due to the hybridization state of N$^3$. Interestingly, the eukaryotic translation system appears to be more stringent against changes within CDSs of mRNAs. Noteworthy, due to the lower amounts of purified translation products from HEK293T cells, we cannot completely exclude the existence of low-level peptide variants (below the detection limit; typically <1%). One could speculate that—among other underlying factors—an increased protein length[49] or the expanded lifespan of eukaryotes[50,51] could require a more accurate decoding in higher organisms.

At the wobble position, a generally less restrictive decoding behavior was observed for both bacteria and eukaryotes, as reported previously[47]. Even nucleotides that could not form an H-bond between N$^1$ and N$^3$ or were completely lacking any H-bonds (i.e., Py and Benz, respectively) provided efficient protein synthesis. In addition, the presence of the 2-amino group within Ap and Dap could compete with the otherwise determining N$^1$–N$^3$ H-bond, confirming a higher structural flexibility at this position. However, decoding of the wobble position also supports the general observation that protein synthesis in eukaryotes is more stringent than in bacteria.

In eukaryotes, A-to-I editing causes a rewiring of the genetic code accompanied by a reduced number of H-bonds during decoding, thereby leading to an amino acid exchange in the respective protein or peptide, as in the cases of GluR-B and 5-HT$_{2C}$R[39–42]. However, this gain in flexibility might come at a cost. The I–C base pair is significantly destabilized in comparison to the standard G–C base pair due to the loss of two H-bonds (one to the C and one to either A1492 or A1493 of the ribosomal A site; E. coli numbering), but this still allows for efficient and accurate translation (Figs. 4 and 5). This is remarkable, since the stability of the codon–anticodon helix at the A site is an important factor for decoding[29]. Indeed, inosine weakens the tRNA–mRNA interactions during decoding, especially if placed at the first or second position of the mRNA codon, as demonstrated using a minimal codon–anticodon model system (Fig. 5a) and by EF–Tu A site binding (Fig. 5b). During elongation, however, a single inosine had no apparent effect, whereas the incorporation of multiple inosines into an mRNA strongly hindered its translation in HEK293T cells (Figs. 4d and 5f). This was unexpected, since it has been reported that up to five editing sites are located in the 5-HT$_{2C}$R mRNA and are efficiently edited in different combinations, depending on the brain areas and the developmental states in which they are expressed[40,52]. In particular, the presence of two inosines within one codon (i.e., concurrent editing of the A- and B-sites), drastically reduced peptide synthesis to levels below the detection limit (Fig. 5f). This indicates that a reduction of the codon–anticodon interaction strength by the inclusion of inosines is only tolerated to a certain extent. Although the absolute editing levels are difficult to assess, strong evidence indicates that multiple inosines are present simultaneously within the 5-HT$_{2C}$R mRNA[40,52].

Over the last decades, different nucleotide analogs and base pairs have been screened for their ability to form stable W–C base pairs, predominately during replication and transcription[53,54]. In line with our findings, most of these nucleotide derivatives provided the structural prerequisites to form at least one H-bond. Interestingly, the pair 6-methoxypurine-thymine forms a central H-bond between N$^1$ (purine) and N$^3$ (pyrimidine) and was reported to have the highest incorporation efficiencies (among the three described base pairs), while the 2-aminopurine-cytosine pair that forms the H-bond between the 2-amino group and the carbonyl oxygen at C$^2$ is less efficiently incorporated into the DNA[54]. The only exceptions are fluorine-containing bases. Although not forming H-bonds, they were incorporated during DNA replication[55,56]. More recently, an artificial base pair was identified that did not depend on the presence of H-bond interactions but still allowed efficient transcription and subsequent translation[31]. In this artificial base pair, the components are highly hydrophobic and their interaction leads to a base pair isosteric to W–C pairs. The missing H-bonds can most likely be compensated by different hydrophobic and stacking interactions[57]. In contrast to these studies, we systematically eliminated potential H-bond partners only from the mRNA codon side in the codon–anticodon helix, revealing the robustness of the decoding process in an authentic setting for protein synthesis. Clearly, changes within purines or pyrimidines impact also polarity, stacking, the syn/anti equilibrium and can lead to steric effects. These parameters contribute to the binding strength in a complex manner and would require

extended and complex quantum–mechanical calculations as well as precise crystallographic structures to provide a satisfactory energy balance of their contributions.

Our study implies that the codon–anticodon helix can accommodate several natural and non-natural base modifications within mRNAs as long as the resulting base pair is isosteric to W–C geometry. The data show that the interactions, in the middle of the base pair, between $N^1$ and $N^3$ at the first two nucleotides are crucial and appear to be sufficient for tRNA binding and for defining of the identity of the codon nucleotide. It should be noted, however, that the ability to decode modified bases might be dependent upon the identity of the codon and on the presence and type of tRNA modifications. In addition, we observe distinct selectivity differences between the bacterial and eukaryotic translation systems with the eukaryotic system appearing less tolerant to nucleotide modifications in the mRNA codons than the bacterial one.

Despite this robustness, the tolerance toward weakening the W–C interactions is limited. In the case of multiple inosines within the CDS, the ribosome is no longer able to provide efficient translation of the modified mRNA. Although A-to-I editing of mRNAs is a potent way to increase the genetic flexibility, this benefit might also come at the cost of losing translation efficiency. It is also conceivable that inosines are deliberately employed to regulate expression by inhibiting protein synthesis. Nonetheless, it is remarkable that the ribosome can tolerate the loss of a variety of interactions between the codon and the anticodon, which illustrates the robust nature of the decoding process.

## Methods

**Sequences**. To generate the template for T7 RNA transcription, a fragment of the eGFP cassette of the lentiviral pHR-DEST-SFFV-eGFP plasmid was PCR-amplified with an N-terminal Flag-tag employing the primers 5′-GCTCTAGA*TAATACGA CTCACTATA*GGGGGGCCACC**ATG**GACTACAAGGACGACGACGATAAGGT GAGCAAGGGCGAGG-3′ (T7 promoter italicized, start codon in bold, and Flag-tag underlined) and 5′-mCmGTCCTCCTTGAAGTCGATGCCCTTCAGCTC-3′. The transcript was then ligated to the respective poly(A)-tailed oligonucleotides yielding the Cap-Flag-eGFP-ErmCL-poly(A), Cap-Flag-eGFP-GluR-B-poly(A), and Cap-Flag-eGFP-5-HT$_{2C}$R-poly(A) mRNAs for assaying the recognition of modified codons HEK293T cells.

**Oligonucleotide synthesis**. Purine, 2,6-diaminopurine, 2-aminopurine, inosine, and ribose-abasic-modified oligonucleotides were purchased from Dharmacon. Zebularine and 2-pyridone-modified oligonucleotides were synthesized in-house[33]. Unmodified oligonucleotides were purchased from IDT.

1-Deazaadenosine and inosine phosphoramidites (for the thermal denaturation studies) were synthesized in analogy to published procedures[58–61]. The synthesis of the linker phosphoramidite ($O^1$-(4,4′-dimethoxytrityl)-1,3-propandiol 3-(2-cyano ethyl-$N,N$-diisopropylphosphoramidite) was performed as previously reported[62]. For the solid-phase syntheses of 1-deazaadenosine or inosine-containing oligonucleotides, either 2′-$O$-tert-butyldimethylsilyl (TBDMS) building blocks with nucleobase tert-butylphenoxyacetyl (tac) protection (ChemGenes, Sigma) or 2′-$O$-triisopropylsilyloxymethyl (TOM) building blocks with nucleobase acetyl protection (ChemGenes, Glen Research) were used. Oligonucleotide synthesis, deprotection, and quality control were carried out as previously described[59,63]. The synthesis of the benzimidazole nucleotide will be published elsewhere.

**_E. coli_ tRNA$^{Gly}$ synthesis**. Standard RNA nucleoside building blocks were used in form of labile base- and 2′-cyanoethoxymethyl (CEM)-protected phosphor-amidites, synthesized according to published procedures[64]. Modified nucleosides were either self-synthesized (DHU, s$^4$U [will be published elsewhere], m$^5$U) or purchased as 2′-TBDMS-protected phosphoramidite (Pseudouridine, Glen Research) (Supplementary Fig. 6).

Deprotection and cleavage was performed under extra mild conditions (2 M NH$_3$ in MeOH, 37 °C, 19 h) to prevent DHU ring opening or substitution products at s$^4$U. Removal of the different 2′-protecting groups was achieved via treatment with 1 M TBAF in anhydrous dimethyl sulfoxide (DMSO) (2 ml) after dissolving the partially liberated RNA in anhydrous DMSO (0.5 ml) and adding 20 µl CH$_3$NO$_2$. After 20 h at 37 °C, the mixture was quenched with 2 ml Tris Buffer solution (Glen Research), and desalted using a HiPrep 26/10 desalting column (GE Healthcare). Purification of the desired RNA was carried out by anion exchange chromatography on a HPLC (Ultimate 3000; Thermo Fisher) on a Dionex DNAPac PA-100 column (22 × 250, Eluent A: 25 mM Tris-HCl, 6 M urea, pH 8.0;

Eluent B: 25 mM Tris-HCl 500 mM sodium perchlorate, 6 M urea) at 80 °C. Product containing fractions were applied to a C18 SepPak catridge (Waters) to remove eluent buffer salts. Elution was achieved with H$_2$O/ACN (1/1, v/v) and the RNA was lyophilized as sodium salt. Final product was identified by anion exchange chromatography on an analytical Dionex DNAPac PA-100 column (4 × 250 mm; eluents as before) and mass spectrometry (7 T FTICR-mass spectrometer; Bruker Daltonics).

**T7 in vitro transcription and poly(A)-tailing**. Capped transcripts encoding the 5′-part of the eGFP reporter mRNAs were generated with the HiScribe™ T7 ARCA mRNA kit (NEB, E2065S). Transcripts without a 5′-cap were synthesized employing the HiScribe™ T7 High Yield RNA synthesis kit (NEB, E2040S) as described by the manufacturer. The oligonucleotides used for assaying translation in HEK293T cells were poly(A)-tailed using the A-Plus™ Poly(A) Polymerase tailing kit (CELLSCRIPT, C-PAP5104H)[33].

**Splinted mRNA ligation**. Two RNA oligonucleotides were ligated to generate the prokaryotic reporter ErmCL mRNA. The oligonucleotide encoding the 5′-part (5′-GGGAGUUUUAUAAGGAGGAAAAAAU**AUG**GGCAUGUUUAGUAUUUU GUAAUCAGCACAGUUC-3′; AUG start codon in bold) and the 3′-part encoding oligonucleotide 5′-P-AUUA<u>U</u>AAACCAAACAAAAAA**UAA**-3′ (The sense codon that was modified or exchanged is underlined; UAA stop codon in bold.) were bridged by a DNA oligonucleotide splinter 5′-TTTGTTTGGTTTATAATGAA CTGTG-3′. The ligation reaction was performed employing T4 DNA ligase (Thermo Fisher Scientific, EL0013)[33]. Ligated full-length mRNA was purified via 8% PAA-urea gels.

Eukaryotic Cap-Flag-eGFP-ErmCL-poly(A) reporter mRNAs were generated by ligating the capped 5′-transcript to the poly(A)-tailed ErmCL oligonucleotide bridged by splinter 5′-TTTTTTGTTTGGTTTATAATCGTCCTCCTTGAAG TCGATG-3′. The enzymatic ligation was performed by T4 RNA ligase 2 (NEB, M0239) as described[33]. Ligation products were purified employing a magnetic mRNA isolation kit (NEB, S1550S). mRNA purity and integrity were checked with a 2100 Bioanalyzer (Agilent). For assaying inosines in a natural mRNA sequence context, the following oligonucleotides were ligated to the capped Flag-eGFP 5′-transcript, GluR-B: 5′-P-AUAUGC<u>A</u>GCAAAACAAAAAA**UAA**-3′ (the A-to-I editing site is underlined; UAA stop codon in bold) and 5-HT$_{2C}$R: 5′-UAGC A<u>A</u>UACGU<u>A</u>AUCCU<u>A</u>UUGAGCAUAGC**UAA**-3′ (the A-to-I editing sites are underlined; UAA stop codon in bold). The ligation sites were bridged with DNA splinters (GluR-B: 5′-TTTTTTGTTTTGCTGCATATCGTCCTCCTTGAAGTC GATG-3′ and 5-HT$_{2C}$R: 5′-CAATAGGATTACGTATTGCTACGTCCTCCTT GAAGTCGATG-3′).

**Prokaryotic in vitro translation**. In vitro translation (IVT) employing the PURExpress Δ ribosome system (NEB, E3313S) was performed as described by the manufacturer[33,65]. Briefly, 1 µM mRNAs were translated in the presence of 10 µCi [$^{35}$S]Met (Hartmann Analytic, SCM-01H) and 1 µM 70S ribosomes. The IVT reactions were incubated for 1 h at 37 °C and were then resolved on Novex 16% Tris-Tricine gels (Thermo Fisher Scientific, EC66955BOX) and exposed to phosphorimager screens, which were scanned using a STORM 840 scanner.

**Cell culture and western blotting**. HEK293T cells (ATCC, CRL-3216) were cultivated in Dulbecco's modified Eagle's medium with 25 mM D-glucose and 4 mM L-glutamine (Gibco, 11965092), 100 U/ml penicillin/streptomycin (Gibco, 15140122), and 10% heat-inactivated fetal bovine serum (Gibco, 10270106). Forty percent confluent HEK293T cells were transfected with 10 pmol of the respective mRNAs using metafectene (Biontex, T020) as described by the manufacturer.

Twenty-four hours after transfection cells were lysed in lysis buffer (20 mM Tris-HCl pH 7.5, 150 mM NaCl, 5 mM EDTA pH 8.0, 1% Triton X-100, Roche complete EDTA-free protease inhibitors). Total protein was quantified via the Bradford assay and 30 µg were separated on Novex 16% Tris-Tricine gels (Thermo Fisher Scientific, EC66955BOX). The resolved proteins were blotted to 0.45 µm PVDF membranes (Amersham, 10600029) employing a Novex XCell II blot module (Thermo Fisher Scientific, 30 min, 100 mA, 20 V). Membranes were blocked with 5% BSA in TBS-T buffer (10 mM Tris-HCl pH 7.4, 150 mM NaCl, and 0.1% Tween-20) for 1 h at room temperature. The blots were probed with an anti-Flag M2 antibody (Sigma, F1804, 1:3000 dilution) or an anti-α tubulin antibody (Abcam, ab4074, 1:7,000) overnight at 4 °C. As a secondary antibody, a goat anti-mouse HRP-conjugated antibody (Dako, P0447) was used in a 1:3000 dilution. The blot was developed using the Pierce ECL western blotting substrate (Thermo Fisher Scientific). Uncropped western blot scans are depicted in Supplementary Fig. 8.

**Mass spectrometry analysis of translation products**. Bacterial translation products were purified employing Vivaspin 2 (5 kDa, Hydrosart, VS02H12) columns and peptides were analyzed as described[33,65]. Flag-eGFP peptides translated in HEK293T cells were purified with anti-Flag M2 magnetic beads (Sigma, M8823). Pulled down proteins were extensively washed with 50 mM ammonium acetate and were directly digested on the beads. Therefore, washed beads were resuspended in ammonium bicarbonate buffer (100 mM, pH 8.0). Proteins were reduced with

dithiothreitol (10 mM) for 30 min at 56 °C, digested for 6 h at 37 °C by adding 0.5 μg Trypsin and alkylated with iodoacetamide (55 mM) at room temperature for 20 min.

Peptides were analyzed using a Dionex, UltiMate 3000 nano-HPLC system (Germering, Germany) coupled via nanospray ionization source to a Thermo Scientific Q Exactive HF mass spectrometer (Vienna, Austria)[66].

Database search was performed using ProteomeDiscoverer (Version 2.1, Thermo Scientific) with search engine Sequest HT. MS/MS spectra were searched against a human protein database (Uniprot, reference proteome, last modified Feb 2018, 20,939 entries) to which 21 different ErmCL protein sequences were added. The following settings were applied: Enzyme for protein cleavage was trypsin; two missed cleavages were allowed. Fixed modification was carbamidomethylcysteine; variable modifications were N-terminal protein acetylation and methionine oxidation. Precursor mass tolerance was set to 10 ppm; fragment mass tolerance was 20 mmu. Maximum false discovery rate (FDR) for proteins and peptides was 1%.

**Thermal denaturation studies**. Absorbance versus temperature profiles were recorded on a Varian Cary 100 spectrophotometer equipped with a multiple cell holder and a Peltier temperature-control device at 250 and 260 nm. RNAs were measured at concentrations ranging from approximately 2 to 8 μM in buffer solutions of 10 mM $Na_2HPO_4$, pH 7.0, containing 150 mM NaCl, while measurements were collected within three complete cooling and heating cycles at a rate of 0.7 °C per minute. Oligonucleotide samples were lyophilized to dryness, dissolved in melting buffer and degassed, and a layer of silicon oil was placed on the surface of the solution to avoid evaporation. Values of $\Delta H0$ and $\Delta S0$ for monomolecular melting transitions were derived from a two-state van't Hoff analysis by fitting the shape of the individual α versus temperature curve[67,68].

**Preparation of ribosomes, tRNAs, and recombinant proteins**. Tight coupled 70S ribosomes used in the EF–Tu binding experiments were isolated from *E. coli* MRE600[69]. Unmodified mRNAs for the EF–Tu-binding experiments (5′-AAG GAGGUAAAAAUGUUUGCU-3′ and 5′-AAGGAGGUAAAAAUGGGGGCU-3′; A site codon underlined) were purchased from Dharmacon. Inosine-modified mRNAs were also obtained from Dharmacon, whereas 2-pyridone-modified mRNAs were synthesized in-house[33].

*E. coli* tRNA[Phe] was purified in-house; tRNA[fMet] was purchased from tRNA Probes[69]. tRNA[Gly] carrying all tRNA modifications was synthesized as described above.

EF–Tu (H84A) was purified employing the IMPACT-CN system (NEB, E6901S)[69]. EF–Tu activity was assessed by native gel assays for tRNA binding (Supplementary Fig. 9)[70]. *E. coli* tRNA nucleotidyl transferase was histidine-tagged and purified. tRNA[Phe] and tRNA[Gly] were [32P]-labeled at the 3′-end using [α-32P]-ATP and *E. coli* tRNA nucleotidyl transferase as described[71]. Aminoacylation of tRNA[Phe] and tRNA[Gly] was performed using purified *E. coli* phenylalanine-tRNA synthetases and glycyl-tRNA synthetase (NEB), respectively[72]. The extent of labeling and aminoacylation was assessed by TLC; the level of aminoacylation was greater than 95%.

**Equilibrium A site binding of EF–Tu ternary complexes**. All filter-binding experiments were performed in buffer A (50 mM Tris-HCl pH 7.5, 70 mM $NH_4Cl$, 30 mM KCl, 15 mM $MgCl_2$, 0.5 mM spermidine, 8 mM putrescine, and 2 mM DTT)[45,71]. In short, activated 70S ribosomes, mRNA (10-fold excess over 70S), and tRNA[fMet] (5-fold excess over 70S) were incubated 30 min at 37 °C to form initiation complexes. Initiation complexes were diluted in buffer A to give a range of concentrations (0.1–1000 nM). In parallel, 3 nM EF–Tu (H84A), 1 mM GTP, 3 mM phosphoenol pyruvate, and 0.25 μg/μl pyruvate kinase were incubated in buffer A for 30 min at 37 °C. In turn, EF–Tu ternary complexes were formed with 0.2 nM 3′ [32P]-labeled Phe-tRNA[Phe]/Gly-tRNA[Gly] for 5 min at 37 °C. Ternary complex reactions were then placed on ice. All subsequent steps were performed with a multichannel pipette. Fifteen microliters of the initiation complex dilutions were transferred to a 96-well conical bottom plate (Nunc) and were mixed with 15 μl of the ternary complex. A site binding was performed for 1 min at room temperature. Twenty-five microliters of the A site-binding reaction mix were filtered through a 96-well filtration apparatus (Schleicher & Schuell) with a nitrocellulose membrane (NitroBind, Maine Manufacturing, 0.45 μm, 1215481) on top of two nylon membranes (EMD Millipore, 0.45 μm, INYC09120)[73]. The membranes were washed three times with 100 μl buffer A and then dried and quantified with a phosphorimager (BioRad). All binding experiments were repeated more than three times. The equilibrium dissociation constant ($K_D$) was determined by fitting the binding data to a one-site binding hyperbolic equation (GraphPad Prism 7).

**Reporting summary**. Further information on research design is available in the Nature Research Reporting Summary linked to this article.

## Data availability

All mass spectrometry data have been deposited to the ProteomeXchange Consortium via the PRIDE database with the data set identifiers PXD011311 (*E. coli* PURExpress translation assay) and PXD011301 (translation in HEK293T cells)[74]. All other data supporting the findings of this study are available within this article and in the Supplementary Information or from the corresponding author upon reasonable request. A reporting summary for this article is available as a Supplementary Information file.

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

## Acknowledgements

We would like to thank Nina Clementi for valuable comments on the manuscript. We also thank the entire Joseph laboratory for providing helpful discussions and a stimulating environment. This work was supported by the Austrian Science Fund (P 22658-B12 and P 28494-BBL to M.D.E., SFB F4411 to A.H., P27947 to R.M., and P30370 to C. K.) and an EMBO short-term fellowship (ASTF 553-2016 to T.P.H.). Funding for the open access charge is provided by the Austrian Science Fund (P 28494-BBL to M.D.E.).

## Author contributions

T.P.H. and M.D.E. designed the experiments. T.P.H., K.F., M.A.J., J.K., C.G., E.F. X.S., and A.S. performed the experiments. T.P.H., H.L., C.K., R.M., S.J., C.H., E.W., A.H., and M.D.E. analyzed the data. T.P.H. and M.D.E. wrote the manuscript. All authors contributed to the production of the final manuscript.

## Additional information

**Competing interests:** The authors declare no competing interests.

