## [Peer Review File · Nature Communications]

Reviewers' comments:

Reviewer #1 (Remarks to the Author):

This is a very interesting work that examines the effects of a number of non-natural base modifications on the decoding process in mRNA translation. Both the decoding potency and the base pair stability of such modifications are thus quantified and this yields new information regarding H-bonding requirements (number and position of H-bonds) for translation. What is nice here is that the emerging picture of how decoding occurs with the modified bases is quite logical, although there are some interesting differences between the bacterial and eukaryotic translation systems. I have only a few comments for the authors to address. On p. 9 it says "incorporation of glutamine instead of arginine", but this should be the other way around. Also, regarding the inosines it says (p. 10) that single inosines did not affect the amino acid composition of the translated product. This sounds weird since the Asn->Ser and Gln->Arg substitutions (Fig. 4E) are examples of this. I don't understand what the authors mean here.

Reviewer #2 (Remarks to the Author):

The authors study the translation efficiency and selectivity of amino acid incorporation using artificial codons containing several types of base analogs by focusing on hydrogen bond interactions between codon and anticodon. The results that amino acid selectivities vary between prokaryotic and eukaryotic systems arouse researchers' interest for the usage of artificial codon and anticodon systems. However, it seems to me that the study is only based on the number of hydrogen bonds, difficult to scrutinize the relationship between translation and codon-anticodon interaction. Since the data are useful for researchers relevant to translation studies, the manuscript should be published in more specific journals, such as *Nucleic Acids Res.*

The strength of hydrogen bonds in each position between pairing bases is different, and thus more quantitative study of hydrogen bonds is required. For example, in replication, the incorporation efficiencies of 6-methoxypruine-thymine, 2-aminopurine- cytosine, and 6-methoxypurine-cytosine are ten-times different among each, despite of each has one hydrogen bond (*Biorg. Med. Chem. Lett.*, 12, 1391, 2002). In addition, for the codon-anticodon interaction, the authors should consider not only hydrogen bonds but also polarity of bases, stacking, and steric effects including hydration for base pairings. Furthermore, the recent study using hydrophobic unnatural base pairs (ref. 44) exhibits that no-hydrogen-bonded base pair can be used for codon-anticodon interaction. However, the authors do not discuss this deeply. There is the contradiction between the authors' idea and the Romesberg's data.

Minor comments:

1. Opposite the abasic site, pyrene might be better than the natural bases as a pairing partner (*Nature*, 399, 704, 1999.). The limitation of the authors' current research is that they use only codon modifications in mRNA. For more precise research, modified anticodons in tRNA should be examined.
2. The authors should discuss the possibility of syn conformation, especially for Benz.
3. In the Fig.1 legend, there is no indication about black and gray color difference of base pairs (probably black for mRNA, and gray for tRNA).
4. Probably, Supplementary Figures S5, S6, and S7 are not cited in the main text

Reviewer #3 (Remarks to the Author):

This manuscript describes experiments designed to test the role of hydrogen bonding in codon-anticodon formation during decoding in the ribosomal A site in bacteria and mammalian cells. The bacterial experiments were performed with an in vitro translational system and the mammalian experiments were done in vivo in human HEK293 cells. The research design involved insertion of non-standard nucleotides into mRNAs that were then translated by normal tRNAs. The experiments clearly show that some hydrogen bonds are necessary for recognition, especially in the first two codon positions; the wobble position was less dependent on hydrogen bonding to form a codon-anticodon interaction and presumably relies on stacking forces, although this could not be tested explicitly in this system. A second part of the manuscript tests the role of inosines edited into certain eukaryotic mRNAs, showing that substitution of inosine for adenine has little effect on how the mRNA is decoded (the amino acids inserted) but the presence of multiple inosines does drastically reduce the rate of translation of the edited mRNA. Overall, this is a very useful set of experiments that for the first time test the roles of specific hydrogen bonds in codon recognition.

The description of the data, however, is somewhat sloppy with the authors claiming that certain non-standard nucleotides are invariably interpreted as equivalent to a single standard nucleotide. This occurs on pages 8-9 and includes claims that zebularine is decoded as C resulting in incorporation of leucine at ZeUU codons (Table S2 and S3 show that isoleucine is incorporated instead of leucine—this could simply be a typing mistake but it occurs in both tables). The authors state that UZeU is partly decoded as Phe but only in the bacterial system despite Table S3 listing Phe as a decoded amino acid in HEK 293. Further on, they state that in the AUP codon, P (purine) is recognized as A whereas in Table S2 the codon is shown specifying Ile or Met showing that P is recognized as A or G. Later, on page 9, they state that DAP (2,6-diaminopurine) in the codon AUDap cannot be decoded as Met in HEK 293 cells despite Table S3 showing a peptide that incorporated Met in response to AUDap.

The authors also make a strong argument for bacteria being “less restrictive towards modified nucleotides” by which, in context, they mean that there are fewer alternative interpretations of the nucleotides (for example, as A or G) in the HEK 293 data than in the bacterial data. That conclusion rests on determining the amino acid sequence of synthesized peptides and the amount of data from the proteins produced in vivo in HEK 293 cells is far less than from the bacterial experiments. The alternative interpretation of the non-standard nucleotides often much weaker, producing much less peptide, so it may be that the alternatives were not observed because of lack of data. The authors should either justify their conclusion or reduce their claim for “less restrictive” translation in bacteria.

Specific comments:

Page 8, line 11 up: “degenerative” should be “degenerate”

Page 9, line 16: “ac4C34” should be “ac2C34”

Page 10, line 13: “ribosomen” should be “ribosome”

Page 12, line 12: the use of “peculiar” here is not needed; although the word can mean “particular or special” it is much more commonly understood to mean “strange or odd”. I doubt that the authors mean that.

Page 12, line 4-5 up: the statement that the eukaryotic system might be more accurate (which hasn't been well determined for the human system used here—the reference is to accuracy in yeast) resulting from longer proteins in eukaryotes is highly speculative. The length of the proteins isn't the

reason eukaryotes might be more accurate but is a adaptationist's explanation for the observation. There are probably many reasons why the situation is as it is but sheer protein length is only one of them and doesn't address the mechanistic reason underlying increased accuracy.

Page 13, line 9 up: "the codon-anticodon helix is not as delicate and fragile as previously expected". No reference is provided to show that this was the expectation and, more importantly, the phrase is far too informal for a paper like this. What, in a technical sense, does "delicate and fragile" mean?

Comments to reviewer #1:

Comment #1: *This is a very interesting work that examines the effects of a number of non-natural base modifications on the decoding process in mRNA translation. Both the decoding potency and the base pair stability of such modifications are thus quantified and this yields new information regarding H-bonding requirements (number and position of H-bonds) for translation. What is nice here is that the emerging picture of how decoding occurs with the modified bases is quite logical, although there are some interesting differences between the bacterial and eukaryotic translation systems. I have only a few comments for the authors to address.*

Response #1: Thank you for the positive assessment of our work. We appreciate your feedback.

Comment #2: *On p. 9 it says "incorporation of glutamine instead of arginine", but this should be the other way around.*

Response #2: Thank you for pointing out this mistake; we changed this part to (p. 10, line 1): “[...] incorporation of **arginine** instead of **glutamine**”.

Comment #3: *Also, regarding the inosines it says (p. 10) that single inosines did not affect the amino acid composition of the translated product. This sounds weird since the Asn->Ser and Gln->Arg substitutions (Fig. 4E) are examples of this. I don't understand what the authors mean here.*

Response #3: We agree that this phrasing was misleading. We formulated the sentence more precise (p. 10, line 5):

“We found that single inosines did not affect the yield of the translated peptide product (Figure 4D). As expected, the inosines were exclusively decoded as G, resulting in an amino acid change from Gln to Arg and from Asn to Ser in the case of the modified 5-HT_{2C}R and GluR-B mRNAs, respectively (Figure 4E and Table S3).”

Comments to reviewer #2:

Comment #1: *The authors study the translation efficiency and selectivity of amino acid incorporation using artificial codons containing several types of base analogs by focusing on hydrogen bond interactions between codon and anticodon. The results that amino acid selectivities vary between prokaryotic and eukaryotic systems arouse researchers' interest for the usage of artificial codon and anticodon systems. However, it seems to me that the study is only based on the number of hydrogen bonds, difficult to scrutinize the relationship between translation and codon-anticodon interaction. Since the data are useful for researchers relevant to translation studies, the manuscript should be published in more specific journals, such as Nucleic Acids Res.*

Response #1: Thank you for reviewing our manuscript. To our knowledge, our study is the first one to systematically address the codon-anticodon interaction during protein synthesis in a natural setting by focusing on artificial codons. Although some limitations are present (see comments below), we think that our data can scrutinize the relationship between the codon-anticodon interaction and decoding. Since decoding is such a central process in every living cell, this manuscript will be of interest for broad readership. As you pointed out, these results might also be taken into account for other biological processes, depending on nucleobase interactions.

Comment #2: *The strength of hydrogen bonds in each position between pairing bases is different, and thus more quantitative study of hydrogen bonds is required. For example, in replication, the incorporation efficiencies of 6-methoxypruine-thymine, 2-aminopurine-cytosine, and 6-methoxypurine-cytosine are ten-times different among each, despite of each has one hydrogen bond (Biorg. Med. Chem. Lett., 12, 1391, 2002).*

Response #2: Thanks for raising a valid point about the number of H-bonds. As you state, the mentioned base pairs 6-methoxypurine-thymine, 2-aminopurine-cytosine, and 6-methoxypurine-cytosine just form one H-bond and show different incorporation efficiencies during replication. However, this seems to be partly explainable by the position of the formed H-bond. The pair 6-methoxypurine-thymine forms a central H-bond between N¹ (purine) and N³ (pyrimidine) and was reported to have the highest incorporation efficiencies (among the three mentioned base pairs). However, 2-aminopurine-cytosine pair, forming the H-bond between the 2-amino group and the carbonyl oxygen at C², is less efficiently incorporated into the DNA (Hirao et al.; 2002). This position dependent effect can be also observed in our study since a single central H-bond only leads to high product yields, if it is formed between the N¹ (purine) and N³ (pyrimidine). In case of pyridone or c¹A, providing the single H-bond at a different location, translation efficiencies were drastically reduced. We changed the manuscript accordingly:

1.) p. 12, line 12: “Translation of the respective codon is only modestly impaired, when H-bonds between the purine-N¹ and the pyrimidine-N³ at the first two codon positions are formed. Thus, in the cases of pyridone or c¹A, the single H-bond is at a different location and translation efficiencies are drastically reduced. The only exception was the translation of single purines within an AAA codon (Lys) in HEK293T cells. [...]”

2.) p. 13-14, line 33: “Over the last decades, different nucleotide analogs and base pairs have been screened for their ability to form stable W-C base pairs, predominately during replication and transcription^{68,69}. In line with our findings, most of these nucleotide derivatives provided the structural prerequisites to form at least one H-bond. Interestingly, the pair 6-methoxypurine-thymine forms a central H-bond between N¹ (purine) and N³ (pyrimidine) and was reported to have the highest incorporation efficiencies (among the three mentioned base pairs), while the 2-aminopurine-cytosine pair that forms the H-bond between the 2-amino group and the carbonyl oxygen at C² is less efficiently incorporated into the DNA⁶⁹. The only exceptions are fluorine-containing bases. Although not forming H-bonds, they were incorporated during DNA replication^{70,71}. More recently, an artificial base pair was identified that did not depend on the presence of H-bond interactions but still allowed efficient transcription and subsequent translation⁴⁴. In this artificial base pair, the components are highly hydrophobic and their interaction leads to a base pair isosteric to W-C pairs. The missing H-bonds can most likely be compensated by different hydrophobic and stacking interactions⁷². In contrast to these studies, we systematically eliminated potential H-bond partners only from the mRNA codon side in the codon-anticodon helix, revealing the robustness of the decoding process in an authentic setting for protein synthesis. Clearly, changes within purines or pyrimidines impact also polarity, stacking, the syn/anti equilibrium and can lead to steric effects. These parameters contribute to the binding strength in a complex manner and would require extended and complex quantum-mechanical calculations as well as precise crystallographic structures to provide a satisfactory energy balance of their contributions.”

Comment #3: *In addition, for the codon-anticodon interaction, the authors should consider not only hydrogen bonds but also polarity of bases, stacking, and steric effects including hydration for base pairings. Furthermore,*

the recent study using hydrophobic unnatural base pairs (ref. 44) exhibits that no-hydrogen-bonded base pair can be used for codon-anticodon interaction. However, the authors do not discuss this deeply. There is the contradiction between the authors' idea and the Romesberg's data.

Response #3: We agree that more than hydrogen bonding is responsible for the formation and stability of Watson-Crick base pairing. We are aware that changes within purines or pyrimidines also impact polarity, stacking, and lead to steric effects. However, an analysis of the translation products does not allow dissecting the exact cause for either lower protein yields or misincorporations. In our opinion even a deeper analysis, lacking extended and complex quantum-mechanical calculations, might not be able to dissect the complex interplay among H-bonds, dipole orientation, stacking, sterics and stereoelectronics. A sentence on this point has been added (p. 14, line 15).

With respect to the mentioned Romesberg's data: indeed, the unnatural base pair in this work (Zhang and Romesberg, 2015) allows the establishment of an intact codon-anticodon interaction. In contrast to our study, the authors screened for a functioning base pair instead of altering only one interaction partner. By combining two novel partners, the lack of H-bonds might be compensated by hydrophobic interactions or other interacting forces. It seems that defining rules for a functioning codon-anticodon base is not trivial, since >200 compounds were screened (Zhang and Romesberg, 2018) to find a suitable couple. In contrast, we aimed at defining and determining the interactions that allow base pairing within a Watson-Crick or Wobble geometry. As you suggested, it would be interesting to systematically alter the tRNA side also in order to find a compensating modification (see comment below). We added a paragraph to the main text to address this and shortly discuss the possibility of base pairing even in complete absence of H-bonds (p. 13, line 33; see comment #2).

Comment #4: *1. Opposite the abasic site, pyrene might be better than the natural bases as a pairing partner (Nature, 399, 704, 1999.). The limitation of the authors' current research is that they use only codon modifications in mRNA. For more precise research, modified anticodons in tRNA should be examined.*

Response #4: We initiated this project to investigate the effects of mRNA modifications on translation. Therefore, we started modulating the codon-anticodon interaction by altering the mRNA codon nucleotides. On the contrary, we do not think this constitutes a limitation because we can study an artificial mRNA in a natural environment. However, we totally agree that altering the helix also by nucleotide derivatives within the anticodon is really interesting. The first chemically synthesized tRNA (interacting with unmodified and modified mRNAs), carrying all natural modifications, was already employed in this study. This is definitely the starting point to extend the investigations on the role of codon-anticodon interactions from "the view" of the tRNAs. However, this is currently beyond the scope of this manuscript.

Comment #5: *The authors should discuss the possibility of syn conformation, especially for Benz.*

Response #5: Indeed, this possibility is difficult to exclude, although it is doubtful it could contribute to isosteric base pairing with natural tRNAs. However, the equilibrium between anti and syn may contribute to the binding efficiency. This is now alluded to in the text (together with comment #2).

Comment #6: *In the Fig.1 legend, there is no indication about black and gray color difference of base pairs (probably black for mRNA, and gray for tRNA).*

Response #6: Thank you for bringing this to our attention. In order to highlight the structures of modified derivatives and to (optically) tone down the standard nucleotides we chose to depict them in black and grey, respectively. We added this information to legend of Fig. 1: “The modified and standard nucleobases are depicted in black and grey, respectively.”

Comment #7: 4. Probably, Supplementary Figures S5, S6, and S7 are not cited in the main text.

Response #7: Thank you. We included the reference to Figures S5 (p. 7, line 17), S6 (p. 10, line 22) and S7 (p. 10, line 24) into the main text.

Comments to reviewer #3:

Comment #1: *This manuscript describes experiments designed to test the role of hydrogen bonding in codon-anticodon formation during decoding in the ribosomal A site in bacteria and mammalian cells. The bacterial experiments were performed with an in vitro translational system and the mammalian experiments were done in vivo in human HEK293 cells. The research design involved insertion of non-standard nucleotides into mRNAs that were then translated by normal tRNAs. The experiments clearly show that some hydrogen bonds are necessary for recognition, especially in the first two codon positions; the wobble position was less dependent on hydrogen bonding to form a codon-anticodon interaction and presumably relies on stacking forces, although this could not be tested explicitly in this system. A second part of the manuscript tests the role of inosines edited into certain eukaryotic mRNAs, showing that substitution of inosine for adenine has little effect on how the mRNA is decoded (the amino acids inserted) but the presence of multiple inosines does drastically reduce the rate of translation of the edited mRNA. Overall, this is a very useful set of experiments that for the first time test the roles of specific hydrogen bonds in codon recognition.*

Response #1: Thank you for those positive comments on our work.

Comment #2: *The description of the data, however, is somewhat sloppy with the authors claiming that certain non-standard nucleotides are invariably interpreted as equivalent to a single standard nucleotide.*

Response #2: We agree that we did not comment on rather rare misincorporation events in the main text. Our intention was not to bloat the manuscript with small effects. We now included them in the main text (see below).

Comment #3: *This occurs on pages 8-9 and includes claims that zebularine is decoded as C resulting in incorporation of leucine at ZeUU codons (Table S2 and S3 show that isoleucine is incorporated instead of leucine—this could simply be a typing mistake but it occurs in both tables).*

Response #3: The comment is absolutely justified. Because MS analysis cannot differentiate between Ile and Leu (because their molecular masses are exactly the same), the software annotated the respective amino acid as Ile. Since the quantitative incorporation of Ile would require a pyrimidine-pyrimidine base pair (Ze-U at the first codon position), the respective amino acid can only be interpreted as Leu. This is in line with our previous work demonstrating that Ze is exclusively interpreted as C in the first codon position. (Hoernes et al., 2018). However, to reduce ambiguity we clarified these entries.

Comment #4: *The authors state that UZeU is partly decoded as Phe but only in the bacterial system despite Table S3 listing Phe as a decoded amino acid in HEK 293.*

Response #4: We included the findings from HEK293T cells into the main text in order to provide a more precise picture. Further, we included a reference to Table S4 that provides the quantities of all identified peptide species (p. 8, line 10).

“In bacteria, UZeU was also partly decoded (~8%) as a phenylalanine (Phe) codon, indicating that Ze can also base pair with A to a limited extent (Figure 2I and Table S2). **Although the base pair Ze-A is also observed in HEK293T cells, it is less abundant than in bacteria (Figure 2J and Table S3 and S4).**”

Comment #5: *Further on, they state that in the AUP codon, P (purine) is recognized as A whereas in Table S2 the codon is shown specifying Ile or Met showing that P is recognized as A or G. Later, on page 9, they state that DAP (2,6-diaminopurine) in the codon AUDap cannot be decoded as Met in HEK 293 cells despite Table S3 showing a peptide that incorporated Met in response to AUDap.*

Response #5: In order to condense the discussion, we did not comment on low-level decoding effects. However, your point is valid. In bacteria, AUP was decoded to ~2% as Met and to ~98% as Ile. In HEK cells, AUDap was decoded to ~98% as Ile and only ~2% as Met. Although we do not discuss these decoding events in detail, we summarize these effects in Table S4. We rephrased the main text not to mislead the readers (p. 9, line 2):

“In accordance with the results obtained when P was within the first two codon nucleotides, this base derivative was decoded **almost exclusively** as an A in the AUP codon (Figure 3C and D, Table S2, S3 and S4).”

Comment #6: *The authors also make a strong argument for bacteria being “less restrictive towards modified nucleotides” by which, in context, they mean that there are fewer alternative interpretations of the nucleotides (for example, as A or G) in the HEK 293 data than in the bacterial data. That conclusion rests on determining the amino acid sequence of synthesized peptides and the amount of data from the proteins produced in vivo in HEK 293 cells is far less than from the bacterial experiments. The alternative interpretation of the non-standard nucleotides often much weaker, producing much less peptide, so it may be that the alternatives were not observed because of lack of data. The authors should either justify their conclusion or reduce their claim for “less restrictive” translation in bacteria.*

Response #6: Thank you for pointing this out and we agree with your comment. Our interpretation rests on the ratio of “miscoded” peptides to “wild-type” peptides, which does not change in dependence of the quantity of the produced peptides. Nevertheless, we cannot exclude that we miss some very low-level variants due to limited amounts produced in HEK293T system. In order to determine the actual detection limit especially for our HEK293T samples in the MS/MS analyses, we calculated the amounts of variants that we would be able to detect within each of our samples by the employed setup. To be able to identify a C-terminal peptide representing a variant with high confidence we require an MS/MS intensity of at least 4.025E+05. By taking the signal of the C-terminal peptides of our samples into account, we could determine the detection limit, varying for each sample. The higher the peptide yield, the better the sensitivity (see below). Thus, we are confident that we could detect low-level variants, supporting our notion that miscoding is more prevalent in the bacterial *in vitro* system than in HEK293T cells. In addition, we commented on this in the discussion (p. 12, line 31):

“Noteworthy, due to the lower amounts of purified translation products from HEK293T cells, we cannot completely exclude the existence of low-level peptide variants (below the detection limit; typically <1%).”

MSMS intensity necessary to identify a C-terminal peptide at peak maximum: 4.02E+05		
Spectrum	MSMS intensity (C-terminal peptide at peak maximum)	Detection limit (based on the MSMS intensity)
P AA	1.45E+08	0.28%
A PA (very low translation activity, see Fig. 2F)	8.72E+06	4.62%
I GG	9.27E+07	0.43%
G IG	2.34E+08	0.17%
A pGG	4.62E+07	0.87%
G ApG	6.97E+07	0.58%
D apGG	8.26E+07	0.49%
G DapG	1.97E+08	0.20%
Z eUU	1.29E+08	0.31%
U ZeU	1.28E+08	0.32%
A UP	6.75E+07	0.60%
A UI	9.36E+07	0.43%
A UAp	9.58E+07	0.42%
A UDap	1.47E+08	0.27%

Comment #7: Page 8, line 11 up: “degenerative” should be “degenerate”

Response #7: We corrected this typo.

Comment #8: Page 9, line 16: “ac4C34” should be “ac2C34”

Response #8: This mistake was corrected.

Comment #9: Page 10, line 13: “ribosomen” should be “ribosome”

Response #9: We revised the sentence.

Comment #10: Page 12, line 12: the use of “peculiar” here is not needed; although the word can mean “particular or special” it is much more commonly understood to mean “strange or odd”. I doubt that the authors mean that.

Response #10: We removed “peculiar” from the sentence and changed it to: “The only exception was the translation of purines within an AAA codon (Lys) in HEK293T cells.”

Comment #11: Page 12, line 4-5 up: the statement that the eukaryotic system might be more accurate (which hasn’t been well determined for the human system used here—the reference is to accuracy in yeast) resulting from longer proteins in eukaryotes is highly speculative. The length of the proteins isn’t the reason eukaryotes might be more accurate but is a adaptationist’s explanation for the observation. There are probably many

reasons why the situation is as it is but sheer protein length is only one of them and doesn't address the mechanistic reason underlying increased accuracy.

Response #11: We agree and adapted this part (p. 12, line 33): “One could speculate that – among other underlying factors – an increased protein length or the expanded lifespan could require a higher translation accuracy in eukaryotes.” (also see comment #6).

Comment #12: *Page 13, line 9 up: “the codon-anticodon helix is not as delicate and fragile as previously expected”. No reference is provided to show that this was the expectation and, more importantly, the phrase is far too informal for a paper like this. What, in a technical sense, does “delicate and fragile” mean?*

Response #12: We rephrased the whole sentence to the following (p. 14, line 21): “Our study implies that the codon-anticodon helix can accommodate several natural and non-natural base modifications within mRNAs as long as the resulting base pair is isosteric to W-C geometry.”

REVIEWERS' COMMENTS:

Reviewer #2 (Remarks to the Author):

The authors revised most part of the manuscript according to all of the referees' comments. They have addressed all of the issues in the revised manuscript. Now the paper should be published.

Reviewer #3 (Remarks to the Author):

I am satisfied with the changes/corrections made to the manuscript by the authors in response to my comments.